# The Life Expectancy Gap between Registered Disabled and Non-Disabled People in Korea from 2004 to 2017

**DOI:** 10.3390/ijerph16142593

**Published:** 2019-07-20

**Authors:** Jinwook Bahk, Hee-Yeon Kang, Young-Ho Khang

**Affiliations:** 1Department of Public Health, Keimyung University, Daegu 42601, Korea; 2Department of Health Policy and Management, Seoul National University College of Medicine, Seoul 03080, Korea; 3Institute of Health Policy and Management, Seoul National University Medical Research Center, Seoul 03080, Korea

**Keywords:** life expectancy, registered disabled people, Republic of Korea

## Abstract

This study aimed to estimate and compare life expectancy at birth among people with and without officially registered disabilities in Korea between 2004 and 2017. We used the National Health Information Database in Korea to obtain aggregate data on the numbers of population and deaths according to calendar year (2004 to 2017), sex, age groups, and officially registered disability status. A total of 697,503,634 subjects and 3,536,778 deaths, including 33,221,916 disabled subjects (829,464 associated deaths), were used to construct life tables. Between 2004 and 2017, life expectancy for people with disabilities increased by 9.1 years in men and 8.3 years in women, while life expectancy for the non-disabled increased by 5.5 years in men and 4.6 years in women. The average life expectancy difference between non-disabled and disabled people was 18.2 years during the study period, decreasing from 20.4 years in 2004 to 16.4 years in 2017. In 2017, the life expectancy of people with the most severe grade of disabilities was 49.7 years, while the life expectancy of people with the least severe grade of disabilities was 77.7 years. The government should implement more effective policies to protect the health of people with officially registered disabilities.

## 1. Introduction

The number of people with disabilities has increased globally in recent decades and is likely to increase substantially with population aging [1]. In South Korea (hereafter “Korea”), registered individuals with disabilities accounted for 2.4% of the total Korean population in 2001, which gradually increased to 4.9% in 2016. Since 2013, more than 40% of the registered disabled people in Korea are aged 65 or older [2]. 

People who have physical or mental disabilities face many health issues. Apart from the disabilities themselves, people with disabilities have a greater prevalence of chronic diseases than non-disabled people, which is attributable to poverty, immobility, poor health behaviors, and the psychosocial distress associated with disabilities [1]. They often have more than one chronic condition at the same time [1,3]. Disabled people also experience poor access to preventive and curative health services and discrimination in the process of healthcare utilization [4,5,6]. Subsequently, people with disabilities have a greater mortality risk than non-disabled people [7,8,9,10]. Investigations into differences in life expectancy between non-disabled and disabled people might enhance public awareness of the health disadvantages faced by disabled people, and could facilitate the implementation of health policies and programs to promote the health of people with disabilities. However, only a few studies have compared life expectancy between people with and without disabilities [11,12]. To the best of our knowledge, no studies have examined life expectancy at birth for people with and without disabilities using total national population data reflecting mortality rates for all age bands starting at age 0. Likewise, time trends of life expectancy for people with and without disabilities have not been explored. We hypothesized that the life expectancy gap between registered disabled and non-disabled people in Korea would vary with calendar years between 2004 and 2017, considering that social factors and healthcare policies have changed during the study periods. This study aimed to estimate and compare life expectancy at birth among people with and without disabilities in Korea between 2004 and 2017 based on whole-population data from the National Health Insurance Service (NHIS).

## 2. Materials and Methods

### 2.1. Study Design, Data, Sample Size

This is a multi-year, cross-sectional analysis on life expectancy using individually linked national mortality follow-up data. This work follows the Strengthening the Reporting of Observational Studies in Epidemiology (STROBE) guidelines for cross-sectional studies [13]. Aggregate data on the numbers of population and deaths according to calendar year (2004 to 2017), sex, age groups (0, 1–4, 5–9, 10–14 …, 85+), and disability status were obtained from the National Health Information Database (NHID) provided by the NHIS in Korea [14]. These data cover all national health insurance beneficiaries in Korea except for foreigners. Appendix A presents annual numbers of population and deaths according to disability status during the study period. Between 2004 and 2017, a summed total of 697,503,634 subjects (33,221,916 people with disabilities and 664,281,718 people without disabilities) and 3,536,778 deaths (829,464 with disabilities and 2,707,314 without disabilities) from each year were analyzed. People with disabilities accounted for 4.8% of the total Korean population between 2004 and 2017. The proportion of disabled people was greater in men than in women. Disabled men accounted for 5.7% of all Korean men, while disabled women accounted for 3.9% of all Korean women during the study period. The proportion of disabled men increased from 4.4% in 2004 to 6.0% in 2017, whereas the proportion of disabled women increased from 2.4% in 2004 to 4.3% in 2017 (see Appendix A).

### 2.2. Registered Disabled Population 

In this study, people with disabilities were defined as those who were officially registered with the Korean government as having a disability. Based on the Welfare of Disabled Persons Act, Korea has registration and grading systems for people with disabilities [15,16]. The Act classifies disabilities into 15 types: physical disabilities, brain lesion disorders, visual impairment, hearing impairment, language disabilities, intellectual disabilities, autistic disorder, mental disabilities, renal impairment, cardiac impairment, respiratory impairment, hepatic impairment, facial disfigurement, intestinal or urinary fistula, and epilepsy disorder. The Act established specific criteria for determining disability grades for each type of disability. The grades range from 1 to 6. Grade 1 is given to a person with the most severe grade of disabilities, and grade 6 to a person with the least severe grade of disabilities. The criteria for each grade for each type of impairment are described in the Welfare of Disabled Persons Act. For example, a person who has lost their arms in the area above the wrist joints, a person who has lost both legs in the area above the knee joints, or a person who cannot move their arms at all due to paralysis would be assigned grade 1. The registration and grading systems provide uniform and standardized health and welfare services to people with disabilities of the same type and grade. In this study, we used information on disability grades, but the disability types were not available for analysis. Appendix A presents the numbers of population and deaths according to calendar year, sex, and disability grades. Grade 1 disabilities accounted for only 0.5% (*N* = 3,462,068) of the total population during the study period, but about 4.9% (*N* = 171,821) of total deaths.

### 2.3. Statistical Analysis

Annual abridged life tables with five-year age groups and a last open interval at age 85 were constructed separately for people with and without disabilities for the years between 2004 and 2017. The Kannisto-Thatcher method was employed to expand the open-ended age interval 85+ to estimate the probability of dying for each of the 5-year age groups 85–89, 90–94, …, 120–124, and 125+ [17]. Additional analyses on time trends of life expectancy differences between non-disabled and registered disabled people and gender differences from 2004 to 2017 were conducted using the least squares regression method (see Appendix A).

### 2.4. Research Ethics 

This study was approved by the National Health Insurance Service of Korea (No. NHIS-2019-1-152) and the Seoul National University Hospital Institutional Review Board ((IRB) No. E-1810-008-975).

## 3. Results

### 3.1. Life Expectancy

Between 2004 and 2017, life expectancy increased in both people with disabilities and non-disabled people, but the magnitude of the increase was greater in disabled people. During the study period, life expectancy for people with disabilities increased by 9.1 years in men and 8.3 years in women, respectively, while life expectancy for the non-disabled increased by 5.5 years in men and 4.6 years in women, respectively (Table 1). 

### 3.2. Life Expectancy Gap

The life expectancy difference between non-disabled and disabled people decreased between 2004 and 2017. The decreased trends were found in both men and women (see Appendix A). The average life expectancy gap among men and women combined was 18.2 years during the study period, and decreased from 20.4 years in 2004 to 16.4 years in 2017. This decline occurred in both men and women. The life expectancy gap between disabled and non-disabled men ranged from 13.8 years to 18.9 years (average of 16.5 years during the study period), while the gap between disabled and non-disabled women ranged from 15.7 years to 21.8 years (average of 18.6 years during the study period). The life expectancy gap between disabled and non-disabled people was larger in women than in men. Between 2004 and 2017, women with disabilities showed higher life expectancies than men with disabilities (Table 1). The difference in life expectancy between disabled men and disabled women was 3.9 years in 2017 (see Appendix A for more detailed results). 

### 3.3. Life Expectancy According to Disability Grades

Figure 1 shows time trends of life expectancy according to disability grades between 2004 and 2017. As the severity of disabilities increased, from grade 6 to 1, life expectancy decreased. This pattern held true for both men and women throughout the study period (see Appendix A for more detailed results). In 2017, the life expectancy of people with the most severe grade of disabilities (grade 1) was 49.7 years, while the life expectancy of people with the least severe grade of disabilities (grade 6) was 77.7 years. The difference in life expectancy between non-disabled and people with disability grade 6 was 6.7 years in 2017. Meanwhile, the average life expectancy gap between non-disabled people and people with disability grade 1 was 35.4 years over the study period, decreasing from 38.6 years in 2004 to 34.6 years in 2017 (see Appendix A for more detailed results). The life expectancy gap between non-disabled people and people with disability grade 1 was greater in women than in men for all calendar years. 

## 4. Discussion

The results of this study indicated that the life expectancy of people with registered disabilities was much shorter than that of non-disabled people. The average life expectancy gap among men and women combined during the study period was 18.2 years. Although disabilities themselves could lead to excess mortality, it is also possible that other factors, such as poverty, socioeconomic disadvantages, poor health behaviors associated with disabling conditions, psychological distress, less social support, and limited access to health care services might also contribute to the greater mortality among those with disabilities than among non-disabled people. The prevalence of chronic diseases has been found to be higher in people with disabilities than in people without disabilities [18]. People with a mental or physical impairment had a higher smoking rate than an age- and sex-matched random sample from the general population [19], although health-related behaviors varied across types of disabilities [20]. Pregnant women with disabilities were found to be more likely to receive inadequate prenatal care than non-disabled pregnant women [21]. Several Korean studies have suggested that people with disabilities are prone to experience poverty, deprivation of human rights, social isolation, and discrimination [22,23,24]. A recent annual report on national disability statistics showed various aspects of socioeconomic disadvantages among disabled people in Korea [2]. In 2017, 58.5% of people with disabilities had an education below the middle school level, while 75.3% of non-disabled people had received a high school education or higher [2]. The proportion of economically active disabled people of working age was 38.9% in 2017, which was 24.9% point lower than that of the total Korean population [2]. More than 30% of households with disabled people were in poverty, as defined by a disposable income below 50% of the national median equivalized income in 2016. This poverty rate was twice the overall poverty rate in Korea [2]. 

The results of this study also revealed that the life expectancy gap was greater in women than in men. The average life expectancy gap between registered disabled and non-disabled people was 16.5 years in men, but 18.6 years in women. The pattern held true for all calendar years examined. The same pattern was also found for the life expectancy gap between non-disabled people and people with grade 1 disabilities. This finding is unexpected, since absolute differences in all-cause mortality according to socioeconomic position are generally greater in men than in women [25]. Prior Korean studies also showed that absolute differences in all-cause mortality and life expectancy according to socioeconomic position indicators, such as education and income, were greater in men than in women [26,27,28]. Two possible explanations can be proposed. First, more severely disabled women (thus having greater mortality risks) might have been registered as disabled in Korea. Prior population survey data from the United States showed that the prevalence and severity of disability in communities were greater in women than in men [29,30]. However, the proportion of the population consisting of disabled people registered with the Korean government in this study was greater in men (5.7% of the total male population) than in women (3.9% of the total female population). Disabled women registered with the government might have had a relatively poor baseline health status compared with disabled men, subsequently leading to a higher risk of mortality among disabled women. Second, Korean women with disabilities are more socioeconomically marginalized than Korean men with disabilities. A recent national report on disabled people indicated huge gaps in educational attainment and economic activities between disabled men and disabled women [2]. In total, 19.0% of disabled women had no formal education, in contrast to 4.0% of disabled men. A total of 57.3% of disabled men had a high school education or higher, while only 29.9% of disabled women did so. This sex difference in education among disabled people is considerable, since the educational gap between men and women in the general population has nearly disappeared in Korea in recent years [26]. In addition, the proportion of economically active individuals among disabled men was 49.5%, while the proportion among disabled women was 29.5% [2].

Over the study period, the life expectancy gap between disabled and non-disabled people decreased in both men and women. A possible explanation for this is that with increases in disability registration, disabled people with better health status might have been more frequently registered in recent years. The proportion of disabled men increased from 4.4% in 2004 to 6.0% in 2017, while the proportion of disabled women increased from 2.4% in 2004 to 4.3% in 2017 (see Appendix A). Another explanation could be positive health effects of the Comprehensive Policy Plans for Persons with Disabilities of the Korean government, which were launched in 1998. According to those policy plans, many progressive policies, including supporting the employment of disabled people, increasing disability benefits, providing free childcare services for disabled children, introducing a disability pension, enacting the Act on the Prohibition of Discrimination against Disabled Persons, and strengthening facilities and tailored services for disabled people have been implemented over the last two decades [2]. 

In this study, the life expectancy gap was found to be especially large in people with severe disabilities. The average life expectancies of people with grade 1 and 2 disabilities in 2017 were 58.9% and 76.4% of the life expectancy of their non-disabled counterparts, respectively. In particular, the life expectancy gap between non-disabled people and people with the most severe grade of disabilities (grade 1) was profound (35.4 years on average over the study period, 38.6 years in 2004, and 34.6 years in 2017). These results remind us that disabled people are not a group of people with homogeneous risk. The life span gradient across disability grades could be a direct result of functional problems due to disabilities themselves [31]. In 2012, the average age at death for people with autistic disorder was at 28.2 years old, while the average age at death for persons with hearing impairment was at 80.3 years old [32]. Disability type and severity might also have influenced access to health care services. Among disabled people, those with severe disabilities showed a lower probability of health screening attendance and medication adherence than those with mild disabilities [18,33,34]. Unmet healthcare needs and medication adherence of people with disabilities differed across types of disabilities [34,35]. 

In addition, the difference in life expectancy between Grade 4 and Grade 5 was minimal until 2010. Then, after 2012, the difference between Grade 4 and Grade 5 began to appear. This might be influenced by the implementation of the mandatory re-assessment policy for disability grades in 2011. This policy requires that persons with disabilities needs to undergo a disability evaluation process that is mandatory after a certain period of time. Furthermore, changes in the criteria for determining disability grades could affect the life expectancy difference between Grade 4 and Grade 5. Between 2004 and 2017, there have been mostly minor changes in the criteria for disability grades, however, major changes occurred in 2010 and 2013. Grade 5 was created for respiratory impairment (which only Grade 1 to 3 existed) in 2010, and for facial disfigurement (which only Grade 2 to 4 existed) in 2013. 

A major strength of this study is that, so far as we are aware, this is the first report of life expectancy at birth and its time trends in registered people with disabilities compared to non-disabled people using complete national population data. However, a limitation of this study is that although we considered the severity of disabilities, we were not able to distinguish among different types of disabilities, despite the possibility that there might be differences in mortality patterns depending on types of disability. For example, some conditions listed in the Welfare of Disabled Persons Act are people with chronic illnesses, such as cardiac impairments, respiratory impairments, hepatic impairments, and intestinal or urinary fistula. These people clearly would have shorter life expectancies. Moreover, people with certain disabilities, such as intellectual disabilities and developmental disabilities, were known to have a lower average life expectancy than the general population [8,36,37]. Therefore, further research should be done to better understand the life expectancy of disabled people according to the type of disability.

## 5. Conclusions

In conclusion, this study showed that people with disabilities had a much lower life expectancy than non-disabled people between 2004 and 2017. This study is significant in that it provides objective and quantitative information on the life expectancy gap between disabled and non-disabled people. The government and society should provide more effective policies to reduce these life expectancy differences and to protect the health of people with disabilities, considering the severity of disabilities.

## Figures and Tables

**Figure 1 ijerph-16-02593-f001:**
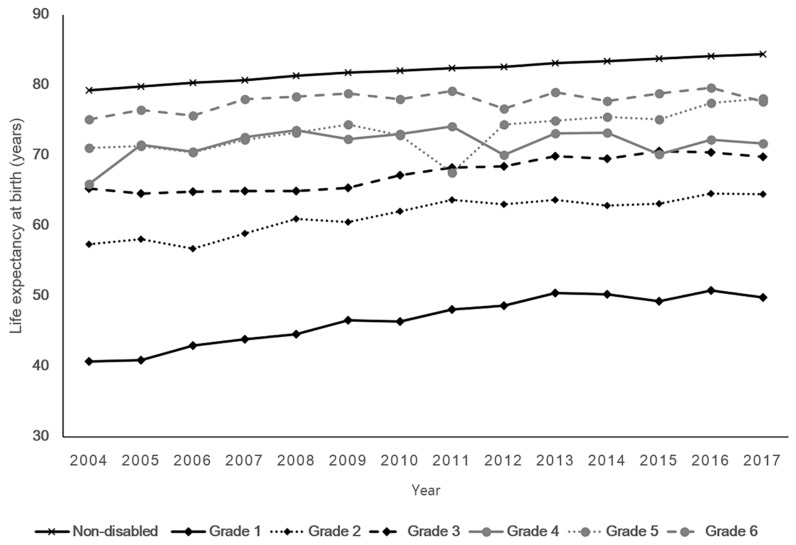
Life expectancy trends between 2004 and 2017 among non-disabled and registered disabled people according to their disability grade.

**Table 1 ijerph-16-02593-t001:** Annual life expectancy among non-disabled and registered disabled people in Korea between 2004 and 2017 by sex.

Year	Men and Women			Men			Women		
	Non-Disabled	Disabled	Difference	Non-Disabled	Disabled	Difference	Non-Disabled	Disabled	Difference
2004	79.3	58.9	20.4	75.8	57.2	18.6	82.2	61.9	20.3
2005	79.8	58.9	20.9	76.5	57.7	18.8	82.5	60.7	21.8
2006	80.3	59.8	20.5	77.0	58.1	18.9	82.9	62.6	20.3
2007	80.7	61.0	19.7	77.3	60.0	17.3	83.3	62.6	20.7
2008	81.3	62.3	19.0	78.0	60.5	17.5	83.9	64.8	19.1
2009	81.8	63.5	18.3	78.4	62.1	16.3	84.3	65.3	19.0
2010	82.0	63.7	18.3	78.6	62.1	16.5	84.6	65.6	19.0
2011	82.4	64.6	17.8	79.0	63.7	15.3	85.0	65.2	19.8
2012	82.6	65.4	17.2	79.2	63.3	15.9	85.1	68.0	17.1
2013	83.1	66.9	16.2	79.8	64.7	15.1	85.6	69.9	15.7
2014	83.4	66.7	16.7	80.2	64.7	15.5	85.9	69.1	16.8
2015	83.8	66.7	17.1	80.6	64.7	15.9	86.2	69.0	17.2
2016	84.2	68.3	15.9	81.0	67.2	13.8	86.6	69.4	17.2
2017	84.4	68.0	16.4	81.3	66.3	15.0	86.8	70.2	16.6

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
