# Peer review of "The Life Expectancy Gap between Registered Disabled and Non-Disabled People in Korea from 2004 to 2017"

_ijerph, 2019, doi:10.3390/ijerph16142593_

Round 1
Reviewer 1 Report
Thanks for the opportunity to review this interesting manuscript entitle "The life expectancy gap between registered disabled and non-disabled people in Korea from 2004 to 2017". There are some issues that should be adressed:
-A clear hypothesis should be adressed before the study main aim.
-Material and methods; The study design as well as the followed criteria (guideliness according to the study design) should be cited and added.
-Material and methods; This section should be widely expanded: study design, population, sample size, outcome measurements and statistical analysis should be accurately detailed.
-Discussion; A limitation subsection should be added before the conclusions. In addition, please add a future studies subsection discussing about intellectual disabilities (Sao Paulo Med J. 2018 Nov-Dec;136(6):505-510. doi: 10.1590/1516-3180.2018.0202161118.), down syndrome (Int J Environ Res Public Health. 2018 May 14;15(5). pii: E983. doi: 10.3390/ijerph15050983.) and others, which may impair quality of life (even related to foot health), but it is necessary to relate these disorders to life expectancy.
-Resutls: Please, summarize your study findings in a short manner.
Author Response
Response to Reviewer 1 Comments
Point 1: A clear hypothesis should be adressed before the study main aim.

Response 1: Based on this comments, we have revised the manuscript including this point (line 47-49 in the revised version).
Point 2: Material and methods; The study design as well as the followed criteria (guideliness according to the study design) should be cited and added.
Response 2: Based on this comments, we have revised the manuscript including this point (line 55-57 in the revised version).
Point 3: Material and methods; This section should be widely expanded: study design, population, sample size, outcome measurements and statistical analysis should be accurately detailed.
Response 3: Based on this comments, we have revised the Material and methods section (line 54-101 in the revised version).
Point 4: Discussion; A limitation subsection should be added before the conclusions. In addition, please add a future studies subsection discussing about intellectual disabilities (Sao Paulo Med J. 2018 Nov-Dec;136(6):505-510. doi: 10.1590/1516-3180.2018.0202161118.), down syndrome (Int J Environ Res Public Health. 2018 May 14;15(5). pii: E983. doi: 10.3390/ijerph15050983.) and others, which may impair quality of life (even related to foot health), but it is necessary to relate these disorders to life expectancy.
Response 4: :Limitations of this study were presented with a strength of this study and further research suggestions before the conclusions (line 226-235 in the revised version). Regarding literatures on intellectual disabilities, we added two references examining life expectancy among people with intellectual disability and developmental disabilities. However, we thought that the papers on foot health among those with intellectual disabilities the reviewer suggested might not be relevant but would be overly specific for this study, considering that this study was on mortality and life expectancy.
Point 5: Resutls: Please, summarize your study findings in a short manner.
Response 5: Based on this comments, we have revised the Results section (line 104-133 in the revised version). To enhance clarity of the message, we divided the Result section into three parts.
Reviewer 2 Report
This paper aims to describe and compare life expectancy at birth among people with and without disabilities in Korea between 2004 and 2017 based on whole population data from the National Health Insurance Service. The data provides useful information for health and social policy to increase service quality for disabled population. Many minor suggestions for this manuscript:
Please describe the effect of disability definition or identification between 2004 to 2014.
To increase the value of this paper, the author can analyze the difference of life expectancy among 16 disability types.
The author can analyze the trend test in table 1 and gender difference.
Figure 1, the author should have a detail discussion of the difference between disability grade 4 & 5.
Author Response
Response to Reviewer 2 Comments
Point 1: Please describe the effect of disability definition or identification between 2004 to 2014.

Response 1: Regarding this point, we have revised the manuscript (line 215-223 in the revised version).
Point 2: To increase the value of this paper, the author can analyze the difference of life expectancy among 16 disability types.
Response 2: When we obtained data from the National Health Information Database, the type of disability was not included. Therefore, it is not possible to analyze life expectancy by type of disability at this moment. Further investigations should be conducted to understand life expectancy according to the type of disability. This point has been documented in the manuscript (line 226-235 in the revised version).
Point 3: The author can analyze the trend test in table 1 and gender difference.
Response 3: Based on the reviewer’s comment, we conducted additional analyses on time trends of life expectancy differences between non-disabled and registered disabled people and gender differences from 2004 to 2017. The decreased trends of life expectancy differences between non-disabled and registered disabled people were significant in both men and women, while gender differences of life expectancy gap between non-disabled and disabled people were not. The results of this analyses were presented in Supplementary Table S3. Regarding the trends, we have revised the Material and methods section (line 96-98 in the revised version) and Results section briefly (line 112-113 in the revised version).
Point 4: Figure 1, the author should have a detail discussion of the difference between disability grade 4 & 5.
Response 4: Regarding this point, we have revised the manuscript (line 215-223 in the revised version).
Round 2
Reviewer 1 Report
Thanks for the opportunity to review this manuscript. Authors have adressed all requeriments and I recommend publication of this manuscript.